# Immunohistochemical Staining Properties of Osteopontin and Ki-67 in Feline Meningiomas

**DOI:** 10.3390/ani14233404

**Published:** 2024-11-26

**Authors:** Gabriele Gradner, Janina Janssen, Anna Oevermann, Alexander Tichy, Stefan Kummer, Stefanie Burger, Ingrid Walter

**Affiliations:** 1Small Animal Surgery Unit, University of Veterinary Medicine, 1210 Veterinaerplatz 1, 1210 Vienna, Austria; janina.janssen@vetmeduni.ac.at; 2Department of Clinical Research and Veterinary Public Health, University of Veterinary Medicine, 3001 Bremgartenstrasse 109 a, 3014 Bern, Switzerland; anna.oevermann@unibe.ch; 3Platform Bioinformatic and Biostatistics, University of Veterinary Medicine, 1210 Veterinaerplatz 1, 1210 Vienna, Austria; alexander.tichy@vetmeduni.ac.at; 4VetCore Facility for Research, University of Veterinary Medicine, 1210 Veterinaerplatz 1, 1210 Vienna, Austria; stefan.kummer@vetmeduni.ac.at (S.K.); stefanie.burger@vetmeduni.ac.at (S.B.); ingrid.walter@vetmeduni.ac.at (I.W.)

**Keywords:** osteopontin, Ki-67, immunohistochemistry, cats, meningioma, WHO

## Abstract

In this study, 53 feline tissue samples from an intracranial tumor called a meningioma were examined for the presence of two specific markers. The Ki-67 index, an indicator of tissue proliferation, and osteopontin, an indicator for proliferation, were evaluated after a standard histopathological grading according to WHO guidelines. High expression of both markers was found, which may explain the high recurrence rate in feline meningiomas.

## 1. Introduction

Meningiomas are tumors originating from the arachnoid cap cells and are the most common intracranial neoplasia in cats. The incidence of feline meningiomas is estimated to be 58.1% of all intracranial neoplasms. Although the majority of meningiomas are microscopically benign, neurological deficiencies secondary to expansive growth and edema can lead to pronounced suffering amongst affected patients [1]. If accessible, surgery is the therapy of choice, as these tumors rarely exhibit invasive growth and complete removal may be curative. However, post-surgical tumor recurrence is reported in 8–22% of cases [1,2,3,4]. Comparably, the estimated five-year recurrence rate after the gross total resection of histologically benign human meningioma varies from 7 to 20% [5]. This unexpectedly high recurrence rate of morphologically benign tumors across species raises the question of a suboptimal estimation of meningioma aggressiveness. Thus, it is of interest to establish further diagnostic tools for the estimation of clinical behavior and the probability of recurrence when evaluating a tumor sample after surgical excision. For this study, pathohistological samples were taken from the archive; so the clinical course, such as survival rate or recurrence rate, could not be described.

Human meningioma grading is standardized by the World Health Organization (WHO) classification system. This grading system relies on histopathological features to subdivide meningiomas into benign (grade I), atypical (grade II) and anaplastic (grade III) tumors. The WHO classification system for domestic animals established in 1999 is outdated [6]. Therefore, veterinary neuropathologists refer to the human classification for grading of small animal meningiomas [4,7,8]. Although the pathological properties and grading of human meningiomas are transferable to small animal tumors, it remains uncertain whether biological behavior can equally be deduced from WHO grading [8,9].

Ki-67 immunohistochemical staining is an established method to interpret the proliferative activity of meningioma [5,10,11,12]. Ki-67 is a non-histone protein that is expressed in all phases except for G0 of the cell cycle. In human medicine, the Ki-67 index is widely used as a prognostic factor for meningioma recurrence [5,13].

Recently, osteopontin (OPN) has become a protein of interest in meningioma research [14]. It is a member of the small integrin-binding ligand N-linked glycoprotein (SIBLING) family and is involved in bone development and remodeling. Additionally, as a secreted protein, it acts as a cytokine regulating tissue repair and inflammation, and takes part in tumor cell proliferation, angiogenesis and metastasis [15]. A study on different types of neoplasia in cats and dogs found a marked osteopontin immunoreaction in several malignant tumors, like mammary adenocarcinomas and soft tissue sarcomas, while weak immunostaining was observed in benign ones [16]. Our group recently published a study on osteopontin expression of meningiomas in dogs, where we found a generally high osteopontin expression across all WHO grades and no correlation with malignancy [17]. To date, there are no publications evaluating osteopontin expression in feline meningioma.

The aim of this study was to evaluate the extent of Ki-67 and osteopontin immunostaining of feline meningioma and to investigate the correlation between Ki-67/osteopontin staining and meningioma subtypes. Based on published data in other species and on recurrence of feline meningioma, we expected a marked expression of these immunohistochemical markers in feline meningioma.

## 2. Materials and Methods

This study involved 53 formalin-fixed (4% neutral formaldehyde, SAV Liquid Production GmbH, Flintsbach a. Inn, Germany) paraffin-embedded (FFPE) blocks of feline meningioma processed in the years from 2001 to 2018 at the Institute of Pathology of Veterinary Medicine in Vienna, Austria. Breed, gender and age were documented. Exclusion criteria were insufficient sample size or quality. Hematoxylin (Thermo Fisher Scientific, Vienna, Austria) and eosin (Carl Roth GmbH & Co. KG, Karlsruhe, Germany) (H&E)-stained tumors were reviewed by a board-certified veterinary pathologist and graded in accordance with the 2016 WHO grading system, which is designed for human tumors [18]. According to the WHO rules, meningiomas were classified as grade I if mitotic rate was <4 per 10 HPF (2.3758 mm^2^) and no brain invasion was observed. Possible subtypes of WHO grade I meningioma were fibroblastic, meningothelial, transitional, psammomatous, microcystic, secretory, angiomatous, lymphoplasmacyte-rich, or metaplastic.

The criteria for grade II tumors included a mitotic rate of 4–19 per 10 HPFs, brain invasion, or three or more of the following histologic features: spontaneous or geographic necrosis, patternless sheets, prominent nucleoli, high cellularity, small cells with high nuclear/cytoplasmic (N:C) ratio. Invasive growth was defined as neoplastic protrusions within the central nervous tissue without pial lining. Possible morphological subtypes were chordoid, clear cell, or atypical meningioma.

Grade III meningiomas were defined as tumors with mitotic rates above 20 per HPF or if they resembled one of the following subtypes: anaplastic, papillary or rhabdoid.

### 2.1. Immunohistochemistry

Briefly, 2.5 µm thick sections were prepared on silanized glass slides and subjected to immunohistochemistry by an indirect immunoperoxidase technique. Endogenous peroxidase activity was blocked with 0.6% hydrogen peroxide in 80% methanol. For the detection of Ki-67, heat retrieval in citrate buffer (pH6) in a steamer was performed. Afterwards, the slides were incubated overnight with the respective primary antibody at 4 °C. Immunohistochemistry for Ki-67 was conducted with a mouse monoclonal anti-Ki-67 (clone MIB-1) antibody 1:1000 (M724029, DAKO, Glostrup, Denmark). For osteopontin, no pretreatment for epitope-retrieval was performed. Immunohistochemistry for osteopontin was executed with a rabbit polyclonal anti-osteopontin antibody 1:300 (PA5-16821, Invitrogen™/Fisher Scientific Scientific, Vienna, Austria). For Ki-67, a secondary anti-mouse and for osteopontin a secondary anti-rabbit (both BrightVision Poly-HRP, ImmunoLogic, Duiven, The Netherlands) antibody were used. The antigen–antibody complex was visualized by using diaminobenzidine (DAB) as chromogen (Quanto, Thermo Fisher Scientific, Vienna, Austria). Counterstaining was carried out with hematoxylin (Mayer’s acidic hemalum solution). All slides were mounted with dibutylphthalate polystyrene xylene (DPX, Sigma-Aldrich/Merck KGaA, Darmstadt, Deutschland). Positive and negative control slides were processed together with the evaluated slides (Ki-67—feline tonsil; osteopontin—feline osteosarcoma and feline kidney). To exclude unspecific binding of the secondary antibody and detection system, in the negative control, the primary antibody was substituted by PBS and unspecific rabbit antiserum IgG (Appendix A).

### 2.2. Slide Scanning and Analysis

For digitalization, a Pannoramic Scan II slidescanner was used (3DHISTECH Ltd., Budapest, Hungary), equipped with a 20× objective.

Image analysis was performed with machine learning algorithms and self-programmed scripts for Ki-67 [19] and osteopontin [20] that were computed using the programs FIJI and ilastik [21,22].

### 2.3. Ki-67

On the digitalized slides, the tumorous tissue was manually identified, and the marked area was analyzed for Ki-67-positive staining with FIJI. Ki-67-positive signal and Ki-67-negative (counterstained) nuclei were separated by color deconvolution with present values for hematoxylin and DAB. Background was automatically reduced with a rolling ball radius of 50. Contrast was enhanced to obtain a normalized signal with similar maximum intensities. Both positive and negative nuclei were segmented and analyzed, excluding very small (<13 µm^2^) and very big (>130 µm^2^) particles to remove staining sprinkles and other artifacts. These filtered particles were counted and set into relation. Ki-67 labeling index (LI) was defined as the percentage of positively stained nuclei in relation to the total number of nuclei.

### 2.4. Osteopontin

As osteopontin is also expressed in the cytoplasm of cells, a calculation of the stained area had to be undertaken. Thus, a combination of machine learning algorithms with ilastik and image analysis with FIJI was used. Ilastik was trained to differentiate between background around the tissue, background in the tissue (vessels and other gaps), osteopontin-positive tissue areas and osteopontin-negative tissue areas. These segmented areas were then passed on to FIJI and area calculation was carried out, considering the osteopontin-positive area to the whole tissue area without background.

For osteopontin, the intensity and extent of staining were both visually evaluated to assess cytoplasmic immunoreactivity. First, the intensity of staining was scored on a scale of 0–3. The score 0 was attained for negative cases, and weak, moderate and strong staining scores were set as 1, 2, and 3, respectively. For the extent of staining, the percentage of positive immunostaining was estimated in each case (OPN positive: OPN negative × 100). The final OPN IHC score was obtained by multiplying the intensity by the percentage, resulting in scores ranging from 0 to 300, with 300 being the highest percentage and the highest intensity.

The Allred evaluation system was the second method for evaluation of osteopontin expression. The Allred score is the sum of the scores from percentage cell staining (no staining = 0, staining of <1% of cells = 1, 1–10% = 2, 10–33% = 3, 33–67% = 4 and 67–100% = 5) and from intensity (absent = 0, weak = 1, moderate = 2, strong =3). Possible scores are 0 and 2–8 [14]. To verify the specificity, osteopontin antibody was tested by Western blotting on lysates of feline osteosarcoma and kidney tissues after migration on a 10% SDS PAGE gel and blotting on a PVDF membrane. Negative controls were represented by the same samples without primary antibody incubation (Appendix A).

### 2.5. Statistics

Statistical analysis was performed with SPSS software (v26 Windows; Chicago, IL, USA). Quantitative data were expressed as means, standard deviation and ranges. Spearman’s rank correlation coefficient (r_S_) was used to calculate the correlations between the OPN IHC Score and the Ki-67 labeling index, and between the Allred Score and the Ki-67 labeling index, which was performed among all samples as well as in each subtype separately. Since the data were not normally distributed, statistical comparison among groups was performed with the nonparametric Kruskal–Wallis test followed by Dunn’s test with Bonferroni’s alpha correction procedure for post hoc comparisons. Dunn’s test was performed with the function dunnTest form the package FSA and using the program R version 4.2.2 [23,24]. For all statistical analyses, a *p*-value below 5% (*p* < 0.05) was seen as significant.

## 3. Results

Western blotting with the rabbit polyclonal antibody (anti-OPN) resulted in only cleaved but not full-length protein bands of osteopontin in two positive controls (Figure 1). Therefore, the antibody might only recognize cleaved osteopontin, at least under reduced Western blotting conditions. (Figure 1)

Fifty-three feline meningiomas were evaluated. In total, 11 samples were biopsies, and 42 samples were retrieved from necropsies. The breeds of affected cats consisted of Domestic Shorthair cats (n = 48; 90.6%), Norwegian Forest cats (n = 2; 3.7%), a Persian cat (n = 1; 1.9%), a Carthusian cat (n = 1; 1.9%) and a Birman cat (n = 1; 1.9%). The gender distribution was as follows: female neutered (n = 19), male neutered (n = 27), and entire female (n = 7), which resulted in an almost equal female to male ratio of 0.96:1. The patients’ ages were in the range of 5–18 years, with a mean of 13 years. Utilizing the 2016 human WHO classification, 50 cases (50/53; 94.3%) were classified as grade I meningioma. Three cases (3/53; 5.7%) were categorized as grade II meningioma as they exhibited signs of brain invasion while otherwise carrying benign histopathological features. Two grade II meningioma were of a mixed fibrous/transitional and one of a fibrous subtype. The following subtypes were identified across grade I and II tumors based on the predominant population: most tumors were found to belong to the fibrous (n = 22; 41.5%) or the transitional (n = 23; 43.4%) subtypes. In eight meningiomas, no predominant features were identified, but these tumors consisted of either a mixed fibrous/transitional (n = 6; 11.3%) or a mixed transitional/meningothelial (n = 2; 3.8%) cell population (Figure 2) (Appendix A).

The overall mean Ki-67 index was 9.19 (range 1.44–47.27). There was no significant difference between the subtypes and the Ki-67 indices (Table 1).

The mean percentage of osteopontin-stained area was 51% (range 1% to 81%). Osteopontin expression levels, assessed by the described methods, were similarly high across the subgroups. A positive correlation between Ki-67 LI and osteopontin expression was observed only in the transitional subtype (Table 2). In total, 32 of the samples exhibited a strong osteopontin staining, 14 showed intermediate staining, and 7 showed weak staining (Figure 3). 

**Figure 3 animals-14-03404-f003:**
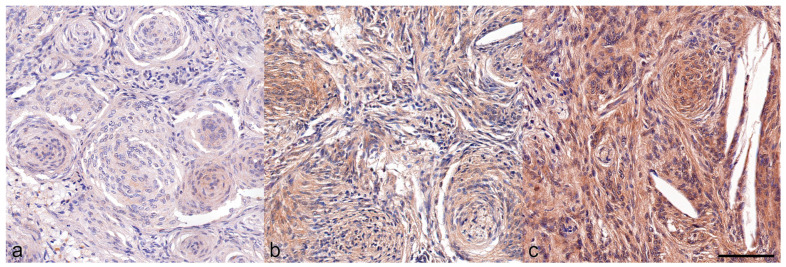
Examples for osteopontin staining scores in feline meningioma. (**a**) Weak. (**b**) Moderate. (**c**) Strong immunolabeling, scale bar = 100 µm.

Twenty-nine meningiomas had osteopontin staining covering 50% or more of the area. The high Allred scores, with a median of 7 (range of 0 to 8), indicated strong staining affinity in the meningioma tissues in our study, supporting the observed high osteopontin expression. Osteopontin staining intensity, staining percentages, Allred scores, and Ki-67 labeling indices did not vary significantly between the subtypes. The sample size in the transitional/meningothelial group (n = 2) was too small for correlation analysis (Table 2).

**Table 2 animals-14-03404-t002:** Comparison of osteopontin expression by IHC score and Allred scoring system and Ki-67 index (% of positive stained nuclei) of subgroups found in feline meningiomas (n = 53).

	Fibrous (n = 22)	Transitional (n = 23)	Fibrous/Transitional (n = 6)	Transitional/Meningothelial (n = 2)
OPN IHC				
Median (min–max)	102.61 (18.0–242.8)	150.17 (0–210.7)	178.02 (57.8–210.0)	85.16 (13.3–157.0)
Mean ± SD	121.46 ± 76.81	115.35 ± 72.41	153.26 ± 61.43	85.16 ± 101.59
Allred scoring				
Median (min–max)	6.0 (4.0–8.0)	7.0 (0–8.0)	7.0 (5.0–8.0)	6.0 (4.0–8.0)
Mean ± SD	6.27 ± 1.32	5.91 ± 1.80	6.83 ± 0.98	6.0 ± 2.82
Ki-67 index				
Median (min–max)	5.16 (1.4–33.9)	8.28 (1.8–47.3)	5.18 (1.5–27.8)	6.7 (6.6–6.8)
Mean ± SD	8.35 ± 9.49	10.21 ± 10.10	9.17 ± 9.69	6.7 ± 0.086
Spearman’s rank correlation Ki-67 LI to OPN IHC	rs = −0.244 *p* = 0.274	rs = 0.485 *p* = 0.022	rs = −0.170 *p* = 0.747	low number
Spearman’s rank correlation Ki-67 LI to Allred score	rs = −0.338 *p* = 0.124	rs = −0.107 *p* = 0.625	rs = −0.018 *p* = 0.973	low number

## 4. Discussion

The aim of this study was to investigate the expression of osteopontin in relation to meningioma grading and Ki-67 staining in cats. In contrast to its benign histopathological appearance, the recurrence rate of feline meningioma has been reported to be up to 20% [2,3]. Therefore, it is necessary to find other prognostic parameters to improve postoperative estimation of recurrence and progression in patients with meningioma.

Based on the WHO grading of 2016, 94.3% (50/53) of the meningiomas in our sample were classified as grade I. This is in accordance with previous studies, as the majority of feline meningiomas carry features of WHO grade I tumors and show very low mitotic activity [4,8]. Commonly, feline meningiomas appear as well delineated and encapsulated masses that rarely infiltrate the brain parenchyma [25]. However, 5.7% (3/53) of tumors in our study were categorized as grade II due to invasive growth. In 2016, brain invasion was added as stand-alone criterion for categorizing a tumor as WHO grade II, when emphasis had previously been placed on morphologic features [18]. The invasive samples in our study were microscopically benign, with two tumors resembling a mixed fibrous/transitional, and one a fibrous subtype. Saito et al. (2021) described five grade II meningiomas (5/45; 2.25%) in their feline sample cohort. They also reported one grade III feline meningioma that exhibited cellular anaplasia and a high mitotic activity of 20 mitoses per 10 HPF, a grade that was not detected in our study [12]. In human meningioma studies, grade II tumors without cellular criteria of malignancy behave similarly to atypical grades regarding recurrence and mortality rate [26]. Based on benign histopathological features alone, there may be an underestimation of the “true” number of grade II meningiomas in cats. Occasionally, brain parenchyma surrounding the tumor may be absent in histopathology samples. Thus, careful sampling and preservation of tissue after surgery is recommended.

As meningiomas are morphologically diverse tumors, 15 subtypes can be allocated according to microscopic appearance [18]. In the current literature, the transitional, the psammomatous, and the fibroblastic subtype are the most common subtypes in cats [27,28,29]. Saito et al. (2021) examined 45 meningiomas and the following subtypes were described: fibrous (15), transitional (22), meningothelial (2), atypical (5) and anaplastic meningioma (1) [12]. This finding fits well with our results, where the transitional and fibrous types were overrepresented. Some of our samples could not clearly be assigned to a special subtype, as the cellular appearance changed on examination of different sample regions. A pattern observed across different kinds of cancers is that WHO grade alone, not the subtype, influences the recurrence- and disease-free interval after surgery [12,30]. The latest update to the WHO classification system was undertaken in 2021, where the 2016 criteria were endorsed for each meningioma grade. New to this edition of the WHO classification was the approval of the use of biomarkers for decision making [31].

Many studies researching human meningioma have inquired into Ki-67 as a factor for tumor grading and prognostic evaluation [5]. A systematic review with meta-analysis investigated the prognostic significance of Ki-67 labeling. Based on the reviewed literature, they determined that a threshold of >4% was associated with an overall poor survival and a reduced disease-free interval [5]. A study on risk factors for recurrence in human WHO grade I meningioma found that a cutoff of Ki-67 > 4.5% had a similar incidence of recurrence in completely resected meningioma, as in those with subtotal resection (18.8 vs. 18.6%) [32]. 

A study on Ki-67 expression in canine WHO grade I meningioma detected positive staining in 91% (n = 70) of samples. However, there was no significant association between Ki-67 staining and outcome [11]. Accordingly, in our recent publication, we found no significant correlation between Ki-67 and WHO grades in canine meningioma graded according to the 2016 human WHO classification [17]. Long et al. (2006) found a significant difference in Ki-67 expression between benign and anaplastic tumors, with benign tumors having a mean Ki-67 LI of 0 (range of 0–2.0), while anaplastic meningiomas expressed a mean Ki-67 LI of 19.3 (range of 13.0–41.1). However, the publication included only 15 meningioma samples and the grading method utilized was the 1999 WHO classification for domestic animals [6,33]. 

In this study conducted on feline meningioma samples, the mean Ki-67 index was 9.19 and there was no significant difference between WHO grades or the subtypes. Saito et al. (2021) also calculated the mean Ki-67 indices for different subtypes of feline meningioma and came to the same results, as neither WHO grades nor subtypes correlated with Ki-67 expression [12]. These findings may indicate that the human WHO grading is not fully applicable to feline meningioma, and that Ki-67 may not be as reliable a prognostic factor as it is in human patients.

Immunohistochemical staining of osteopontin in our cohort of feline meningioma samples revealed intense tissue staining combined with a high ratio of positively stained areas. This is the first study investigating osteopontin in feline meningioma tissues. Shigeyama et al. (1996) examined the role of osteopontin in feline teeth and found intense immunostaining in regions of osteoclastic activity [34]. The authors successfully used a human osteopontin antibody on feline samples. We utilized a rabbit polyclonal anti-osteotpontin antibody, which has been used in previously published studies on canine meningioma tissue [17]. We copiously tested it for binding specificity on feline tissues and conducted Western blotting as a second method of validation and found that splice variants or cleaved protein fragments smaller than the full-length protein were detected. Together with the similarity of immunostaining in human and feline kidneys, and expected results in negative controls, we assume the antibody to be specific. The functional domains of osteopontin are highly conserved among species [33]; however, sequence similarity between human and feline species is below 70%. However, a limitation due to the possibility that not all osteopontin protein forms were recognized by the antibody in feline tissue should be kept in mind. As the functional domains of osteopontin are highly conserved among species, good inter-species binding of antibodies against osteopontin is not surprising [35]. 

Several studies on human neoplasia have investigated the expression of osteopontin in various malignancies, like breast and prostate cancer, squamous cell carcinoma, melanoma, osteosarcoma and glioblastoma [36]. However, results in different tumor types vary. Zhou et al. (2005) found an increased expression of osteopontin in metastatic melanomas, but no correlation regarding survival time or metastasis [37]. In contrast, Rangel et al. (2008) reported that higher osteopontin staining in melanomas correlated with a decrease in recurrence-free intervals and disease-specific survivals [38]. Klopfleisch et al. (2010) found no statistically significant difference in osteopontin staining between canine mammary carcinomas and adenomas, whereas a study conducted on a variety of canine and feline tumors found a strong expression of osteopontin in morphologically malignant tumors [16,39].

According to a study investigating osteopontin expression in human meningioma performed by Li et al. (2018), there was a significant increase in osteopontin expression with histological grade and recurrence. Furthermore, the protein seemed to play an important role in the development and progression of meningioma [40]. Osteopontin was also found to be a valuable marker for the prediction of short-term recurrence in human WHO grade I meningioma. In the study published by Tseng et al. (2010), 9 out of 32 patients (28%) with a grade I meningioma experienced a recurrence within a mean follow-up time of 34 months (5–65 months). In recurring tumors, osteopontin staining was found to be approximately 6 times higher as compared to non-recurring tumors. An osteopontin Allred score between 0 and 3 was associated with a recurrence-free time of more than 25 months. In comparison, an osteopontin Allred score from 4 to 8 was indicative of a shorter average recurrence-free time [41]. Osteopontin can be utilized to distinguish between WHO grades I and II and to predict recurrence [42]. 

The time to recurrence of a resected WHO grade I meningioma may be longer than the residual life span of a cat, as reported post-surgical median survival times range from 23 to 37 months [1,4]. However, in small animal case series, the true recurrence rate is hard to determine because of missing diagnostic imaging or the loss of patients to follow-up. Due to the retrospective nature of this study, no statement regarding prognosis or survival could be made. One significant limitation of this study is that we cannot correlate the expression markers with the recurrence rate or survival times. This means that the research results are not yet sufficiently validated to make a definitive statement about their significance in clinical application. Therefore, additional studies are required to ensure their relevance and applicability in the clinical context. It is also important to bear in mind that it is difficult to achieve free margins of the tumor in intracranial surgery. The likelihood that an intracranial tumor will relapse should therefore be considered to be higher than where it can be removed with large margins.

Transferring the published knowledge to our findings of immunohistochemical staining, the overall high expressions of OPN and Ki-67 may implicate a tendency towards clinical aggressiveness. These findings may contribute to the high rates of recurrence that have previously been reported for microscopically benign feline WHO grade I meningioma.

## 5. Conclusions

In conclusion, the expressions of OPN and Ki-67 were generally high but not statistically different in feline meningioma subtypes. Furthermore, a prevalence of 5.7% for WHO grade II meningioma in cats was detected when utilizing the human WHO classification published in 2016. The intensity and extent of OPN and Ki-67 staining in cats may hint at the underlying mechanisms contributing to the higher tendency of recurrence in these otherwise benign tumors.

## Figures and Tables

**Figure 1 animals-14-03404-f001:**
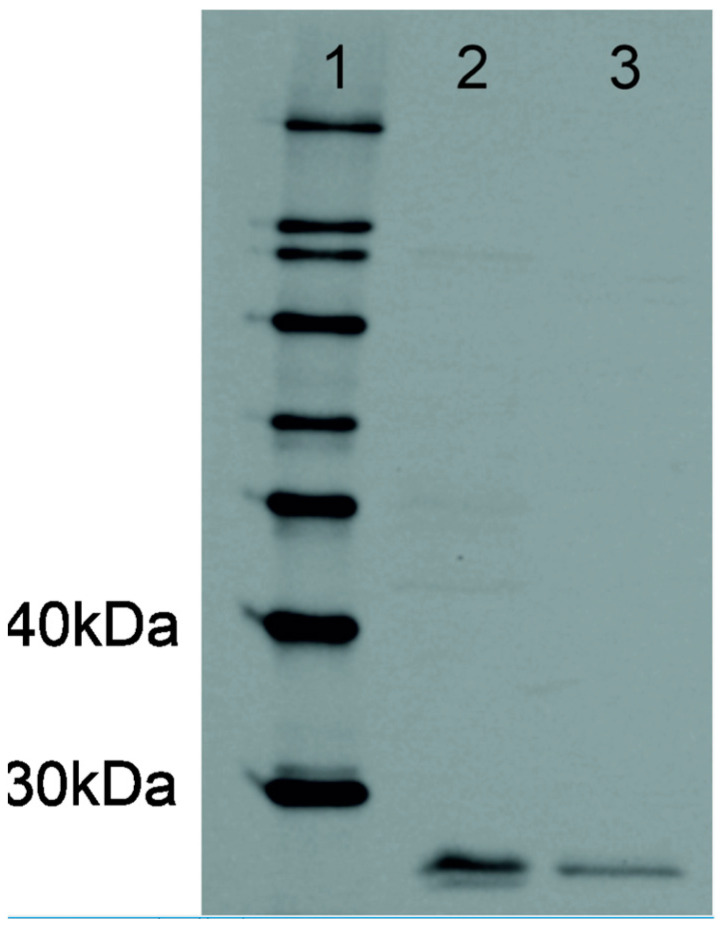
Western blotting of OPN. Lane 1: molecular weight marker; lane 2: kidney, cat, SDS buffer; lane 3: osteosarcoma, cat, Tris Triton buffer. The resulting bands represent the cleaved OPN.

**Figure 2 animals-14-03404-f002:**
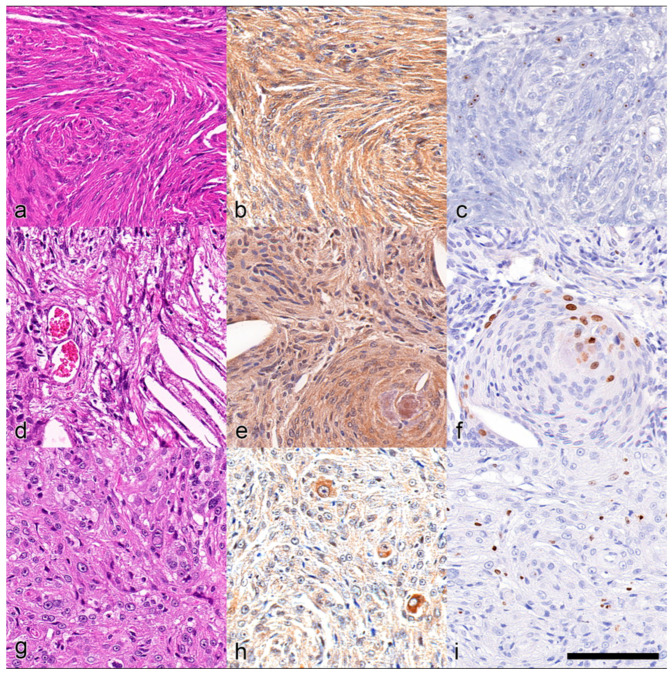
Immunohistochemical staining of osteopontin and Ki-67 expression in different meningioma subgroups. (**a**) Grade I meningioma of the fibrous subtype. (**b**) Osteopontin expression. (**c**) Ki-67 staining. (**d**) Grade I transitional subtype with cholesterol clefts. (**e**) Osteopontin expression in the transitional subtype. (**f**) Ki-67 in the transitional subtype. (**g**) Grade I meningothelial meningioma. (**h**) Osteopontin expression. (**i**) Ki-67 positive staining of nuclei in the proliferative stage. Scale bar = 100 µm.

**Table 1 animals-14-03404-t001:** Comparison of osteopontin expression by IHC score and Allred scoring system and Ki-67 index (% of positive stained nuclei).

	OPN IHC	Allred Scoring	% Ki-67 Index
Median (min–max)	150.17 (0–242.8)	7 (0–8)	6.14 (1.4–47.3)
Mean ± SD	121.04 ± 73.12	6.17 ± 1.55	9.19 ± 9.47

## Data Availability

All data generated or analyzed during this study are included in this published article.

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
