# Peer review of "Immunohistochemical Staining Properties of Osteopontin and Ki-67 in Feline Meningiomas"

_animals, 2024, doi:10.3390/ani14233404_

Round 1

Reviewer 1 Report

Comments and Suggestions for Authors

This manuscript investigates the expression of Osteopontin (OPN) and Ki-67 in feline meningiomas to explore their potential as markers of clinical aggressiveness and recurrence. The study is well-motivated, considering the recurrence rate (8-22%) of these tumors despite their benign appearance. However, there are several critical methodological concerns and areas where the manuscript could be improved to enhance clarity, validity, and scientific rigor.

Major Points:

  1. Study Design Limitation Regarding Clinical Outcomes:

The primary aim stated in the introduction is to establish diagnostic tools for estimating clinical behavior and the probability of recurrence in feline meningiomas (line 48-50). However, the study only compares Ki-67 and OPN expression with histological grades and subtypes, without correlating these markers with actual clinical outcomes such as recurrence rates or survival data. This is a fundamental limitation of the study design. To identify reliable prognostic markers for recurrence or clinical aggressiveness, it is essential to correlate Ki-67 and OPN expression with clinical follow-up data. The authors should either adjust the stated aim of the study to reflect the scope of their analysis or, if possible, include clinical outcome data to strengthen the relevance of their findings. At the very least, the introduction should acknowledge this limitation upfront, and the discussion should critically evaluate how this limitation affects the interpretation and applicability of the results.

  1. Antibody Specificity and Western Blotting:

The authors performed Western blotting to confirm the specificity of the antibody used for OPN detection. However, they observed a band below 30 kDa, which does not correspond to the expected molecular weight of OPN. Other studies using the same antibody in human tissue report OPN at approximately 52 kDa. The predicted molecular weight by sequence is around 33 kDa, with higher bands (44–66 kDa) due to post-translational modifications. The authors should clarify whether the observed band corresponds to the expected molecular size of OPN. If the antibody primarily detects the cleaved form of OPN rather than the full-length protein, this limitation should be acknowledged. Also, appropriate negative controls should be included to demonstrate that the observed band is specific and not due to background noise. The rationale for using kidney tissue as a positive control should be explained, considering that OPN is upregulated in renal injury. Did authors used injured feline kidney for western blot? The results of the Western blot should be presented in the Results section, not just in the Methods.

  1. Interpretation of OPN Expression:

Using an antibody that only recognizes the cleaved form of OPN may limit the study's conclusions about OPN's role in feline meningiomas. Discuss the implications of detecting only the cleaved form of OPN. Consider using or validating an antibody that recognizes both full-length and cleaved forms to provide a comprehensive analysis of OPN expression. This approach would offer more accurate insights into the protein's role in tumor behavior and potential as a prognostic marker.

  1. Statistical Analysis Clarity (Line 158):

The statistical analysis section lacks clarity regarding the use of the Kruskal-Wallis and Mann-Whitney tests. The statement "for post hoc comparisons, Mann-Whitney tests were performed" is potentially misleading because the Mann-Whitney U test is not a post hoc test following Kruskal-Wallis. Please clarify the statistical methods used. If the Kruskal-Wallis test was employed to detect differences among three or more groups, appropriate post hoc tests like Dunn's test with Bonferroni correction should be used. If only two groups were compared, the Mann-Whitney U test suffices without the need for Kruskal-Wallis. Ensure that the statistical methods align with the data structure and the comparisons made.

  1. Presentation of p-values in Table 2:

It is unclear whether the p-values in Table 2 result from the statistical tests mentioned in the text, as only "Spearman rank correlation" is indicated. Specify in the table caption or footnotes which statistical tests were used to generate the p-values. Clearly distinguish between correlation analyses and group comparisons to improve transparency.

  1. Inclusion of IHC Control Images:

The manuscript lacks images of the negative and positive controls for the IHC experiments. Include images of the IHC negative and positive controls in the supplementary material. This addition will help validate the specificity and effectiveness of the staining protocols used.

  1. Quality of Figure 2 Images:

The current images in Figure 2 may not be at a high enough magnification to clearly show the nuclear expression of Ki-67. Provide higher magnification images where the Ki-67 nuclear staining is clearly visible. This enhancement will improve the visual representation of the results and support the findings presented.

Minor Points:

  1. Clarification of WHO Grading System (Line 95):

The manuscript mentions the use of the "2016 WHO grading system" without immediate clarification or citation. Immediately cite the relevant reference (e.g., reference 18) after mentioning the 2016 WHO grading system. Specify that this grading system is designed for human tumors to avoid ambiguity.

  1. Supplementary Material Placement:

The supplementary material appears to be included within the main manuscript text. Remove the supplementary material from the main text and submit it as a separate file, as per the journal's guidelines.

  1. Formatting Inconsistencies:

There are inconsistencies in font style and size throughout the manuscript, and some text appears in blue. Review and correct the manuscript formatting to ensure consistency with the journal's style guidelines. Remove any colored text unless specifically required.

  1. Reference Style Compliance:

The reference style does not fully comply with the journal's format. For example, in reference 2, the journal name abbreviation is incorrect. Reformat all references to match the journal's required citation style. Pay attention to journal name abbreviations, punctuation, and formatting details.

Reviewer 2 Report

Comments and Suggestions for Authors

Dear authors, 

the present manuscript entitled “Immunohistochemical Staining Properties of Osteopontin and  ki-67 in Feline Meningiomas” describes the potential use of osteopontin as marker in this disease.

If we consider the low incidence of this tumour in feline species the authors analysed an important number of cases, and this represents a merit of the present research.

The methodology based exclusively on IHC to evaluate ki-67 and osteopontin expression is not highly innovative although the experimental design is well conducted, and the size sampling is significant. Material and methods are clear and complete and the discussion is well argued.

However, the results and the figures description  to be improved as follows:

Change OPN with Osteopontin throughout the entire text and figures and tables. It sounds better than OSN.

1)       Figure 1. Western Blot OPN. Please insert this figure in the results section. Moreover did the authors perform also tubulin staining as well beta actin as housekeeping? It should be inserted in the figure. Did the authors test osteopontine antibody also on positive human control?

2)      Figure 2. Change the legend as follows

Immunohistochemical staining of Osteopontine and Ki-67 expression in different meningioma sub-groups.

a)       Grade 1 meningioma of the fibrous subtype (H&E x 200). b) Osteopontine expression (IHC x 200). C) Ki-67 staining d Grade 1 transitional subtype with cholesterol clefts (H&E x 200). e) Osteopontine expression in the transitional subtype. f) Ki-67 positivity in grade I meningioma. g) Grade 1 meningothelial meningioma (H&E x 200). h) Osteopontine expression (IHC x 200).

I) Ki-67 positive staining of nuclei in the proliferative stage.

b)      Can the authors improve the quality and the contrast of all the figures? Additionally, Ki 67 nuclear staining is not evident. A 400X magnification of Ki67 staining should be more evident.

3)      Figure 3

Please improve the quality of figure

4)      TABLE1 and TABLE 2

Chek the legend text. There are typing error and rephrase the legend text of the tables.

Additionally

In table 1 change the columns as follows

         OPN IHC with Osteopontine expression

         Allred with Allred scoring system

         % Ki-67 ratio with % Ki-67 positivity

Change the title of the table 1 as follows:

Comparison of Osteopontin expression by IHC score and Allred scoring system and Ki-67 index

 (% of positive stained nuclei).

Rephrase the legend according to these changes

TABLE 2  

Rephrase the legend of the table

Chek again the Journal Instruction Guidelines for tables and Figures.

Reviewer 3 Report

Comments and Suggestions for Authors

Summary:

The authors reviewed tissues from 53 previously diagnosed meningioma samples from cats. After reviewing histology and grading the tumors according to current WHO guidelines, immunohistochemistry (IHC) was performed for Ki-67 and osteopontin (OPN). These markers were chosen due to their prognostic significance in the same or similar tumors in human medicine. Ki-67 was evaluated via established indexing protocols. OPN was similarly evaluated by two established IHC scoring systems. The vast majority of the meningioma samples were histologically benign according to current WHO guidelines, but the expression of Ki-67 and OPN were high. No correlations were found between the expression of the markers and the histologic features of the tumors. The authors discuss possible causes and implications of this apparent discrepancy, including discussion that the WHO guidelines, which were established for human tumors, may not be accurate in predicting the behavior of meningiomas in feline patients.

Review and comments:

This paper helps expand our understanding of this common tumor in cats, as well as tumor biology as a whole. Although significant correlations that offer objective diagnostic criteria were not found, investigating these potential IHC markers is essential. The paper demonstrates areas where additional research is needed and further reinforces the need for revised histopathological grading systems for feline meningiomas.

The authors’ methods are scientifically sound, and the reviewer appreciates the highly detailed descriptions in the Materials and Methods section.

In the Discussion section, the authors do a good job of contextualizing their findings within the current literature and comparing their results to other studies of feline meningiomas.

The figures and tables are accurately labeled and clear to the reader.

The cited references are appropriate for the study.

The paper is overall very well-written and suitable for publication.

The authors rightfully place a high degree of importance on the expression of Ki-67 and OPN as possible causes for recurrence, but other possible factors should be elaborated on as well. For example, are surgeons able to achieve adequate excisional margins, and if so, by how many millimeters? I imagine that the tumors are not removed with wide margins due to the nature of intracranial surgery, as such some neoplastic cells may remain to promote recurrence. How often do the pathologists’ reports on histological margins agree with the surgeon regarding the perceived gross margins? I would like the authors to mention these factors, even if only briefly.

The supplementary table on pages 10-14 is difficult to read and should be reoriented and resized. This may be a job for the editing team of the journal rather than the authors, though.

The use of English language is overall excellent with very few errors, as detailed below.

Detailed corrections:

Line 17: should be “according to WHO guidelines”

Line 39: Reporting an “occurrence of 58.1%” could be misinterpreted as saying that 58.1% of cats will develop meningiomas. This sentence should be re-written to clarify that the incidence of meningiomas is estimated to be 58.1% of all intracranial neoplasms in cats.

Line 49: A comma is not needed after “recurrence” in this sentence.

Line 58: Similarly, no comma is needed after “uncertain” in this sentence.

Line 63, 106, 107, 109, 122, 136, 139, 140, 142: “Osteopontin” is not a proper noun, and in the English language should not be capitalized unless at the start of a sentence.

Line 113: A period should be added after the parenthesis.

Round 2

Reviewer 1 Report

Comments and Suggestions for Authors

The authors have adequately addressed all the raised concerns and have made the necessary corrections appropriately. Additionally, they have provided particularly suitable negative controls to thoroughly demonstrate the specificity of the antibody, further strengthening the validity of their results. The revised manuscript meets the required standards and is ready for publication.

Author Response

Dear Reviewer, 

thank you for reviewing the paper. 

Kind regards, the authors

Reviewer 2 Report

Comments and Suggestions for Authors

Dear Authors,

Thank you for answering to all my comments. I retain that this manuscript is now acceptable in thi journal.

However there are some typing errors troughtthe manuscript

line 26 of Abstract The aim of this study is not related to methods section plaese correct. each section the abstract ( introduction , material end methods etc) need to be inserted with double points. Plaese chek all the abstract according to jourla style.

 line 127 change with "To exclude unspecific binding of the secondary antibody and detection system, in negative control the primary antibody was substituted by PBS and unspecific rabbit antiserum IgG (Figure S1).

line 167: change with To verify specificity, osteopontin antibody
was tested by Western Blot on lysates of feline osteosarcoma and kidney tissues after migration on a 10% SDS PAGE gel and blotting on a PVDF membrane.

line 169
Negative control were represented by the same samples  without primary antibody incubation. (Figure S2).

Author Response

Dear Reviewers, 

thank you for thoroughly reviewing our paper. We have revised the style of the abstract to align with the journal´s guidelines, ensuring it is now a single paragraph without headings. 

Yours sincerely, 

The authors

line 26 of Abstract The aim of this study is not related to methods section plaese correct. each section the abstract ( introduction , material end methods etc) need to be inserted with double points. Plaese chek all the abstract according to jourla style.  Thank you-changed according to the guidelines. 

 line 127 change with "To exclude unspecific binding of the secondary antibody and detection system, in negative control the primary antibody was substituted by PBS and unspecific rabbit antiserum IgG (Figure S1).  Thank you - inserted. 

line 167: change with To verify specificity, osteopontin antibody
was tested by Western Blot on lysates of feline osteosarcoma and kidney tissues after migration on a 10% SDS PAGE gel and blotting on a PVDF membrane. Thank you - changed. 

line 169
Negative control were represented by the same samples  without primary antibody incubation. (Figure S2). Thank you - changed
